# Lenghu on the Tibetan Plateau as an astronomical observing site

Licai Deng[1,2,3 ✉], Fan Yang[1,2 ✉], Xiaodian Chen[1,2,3 ✉], Fei He[4,5 ✉], Qili Liu[6], Bo Zhang[1], Chunguang Zhang[1,2,3], Kun Wang[2], Nian Liu[2], Anbing Ren[2], Zhiquan Luo[2], Zhengzhou Yan[2], Jianfeng Tian[1] & Jun Pan[1]

On Earth's surface, there are only a handful of high-quality astronomical sites that meet the requirements for very large next-generation facilities. In the context of scientific opportunities in time-domain astronomy, a good site on the Tibetan Plateau will bridge the longitudinal gap between the known best sites[1,2] (all in the Western Hemisphere). The Tibetan Plateau is the highest plateau on Earth, with an average elevation of over 4,000 metres, and thus potentially provides very good opportunities for astronomy and particle astrophysics[3–5]. Here we report the results of three years of monitoring of testing an area at a local summit on Saishiteng Mountain near Lenghu Town in Qinghai Province. The altitudes of the potential locations are between 4,200 and 4,500 metres. An area of over 100,000 square kilometres surrounding Lenghu Town has a lower altitude of below 3,000 metres, with an extremely arid climate and unusually clear local sky (day and night)[6]. Of the nights at the site, 70 per cent have clear, photometric conditions, with a median seeing of 0.75 arcseconds. The median night temperature variation is only 2.4 degrees Celsius, indicating very stable local surface air. The precipitable water vapour is lower than 2 millimetres for 55 per cent of the night.

The geographic information of the site, Lenghu in Qinghai Province, is summarized in Methods and Extended Data Fig. 1. The main site parameters—including cloudiness and night-sky background brightness, air temperature, pressure, humidity, wind speed and direction, dust, precipitable water vapour (PWV), and, most importantly, seeing (using a differential image motion monitor (DIMM)[7,8]—have been monitored starting at different times from March 2018 onwards (summarized in Extended Data Table 1). As DIMM seeing must be measured in the vicinity of a telescope project and at a similar height from the ground as the telescope, a 10-m tower was built to mount the DIMM. Shortly after the initial site reconnaissance, to start the site monitoring as soon as possible, the building materials and tools were carried to the site by a helicopter and the scientific devices were manually carried up to the mountain in September 2018, before the road reached the site. This could not have been accomplished without the great assistance from the local government of Lenghu Town. All the measurements and preliminary statistics of the raw data are updated daily and are available at http://lenghu.china-vo.org/index.html. Comprehensive comparisons of the key site characteristics of Lenghu with those of the other known best astronomical sites in the world are summarized in Table 1. A detailed analysis is given in the following.

## Available observing time

For any modern observatories for night optical/infrared astronomy and planetary sciences[9], the first factor to consider is undoubtedly the clarity of nights, followed by the darkness of the night sky (that

is, avoiding light pollution), among other parameters important for advanced modes of observations. Light pollution is mainly the result of human activities. Qinghai Province on the Tibetan Plateau has a very low population; therefore, problematic artificial light sources are at present non-existent. However, this does not mean that industrial development will not occur in the future. If the local population were to grow with economic development, then control of light pollution could be lost. This potential conflict between scientific research and industry needs a resolution[10]. Owing to the enforceable and long-term night-sky protection policy issued by the local municipal government in 2017, a priori, such a potential threat to astronomical observations has been lifted. Night-sky protection in the whole area of Lenghu will be guaranteed by law.

First, we determined how dark the local night sky is. We monitored the night-sky brightness using a widely used commercial sky quality meter (SQM)[11], which has a wide passband from 400 nm to 600 nm centred at the Johnson V-band and accurately measures the integrated light of the entire visible sky, with the sensitivity optimized towards the zenith and a quick drop-off to less than 20% once the zenith angle is greater than 60°. The integrated full visible sky brightness is converted into zenith brightness in mag arcsec$^{-2}$ (ref. [12]). The night-sky brightness reaches 22.3 mag arcsec$^{-2}$ during a fully clear new moon time, in the extreme case when the bright part of the Galactic Disk is far away from the local zenith. The average night-sky brightness is around 22.0 mag arcsec$^{-2}$ when the Moon is below the horizon, comparable to the other three sites in Table 1. Artificial light contributions are completely negligible.

[1]CAS Key Laboratory of Optical Astronomy, National Astronomical Observatories, Chinese Academy of Sciences, Beijing, China. [2]Department of Astronomy, China West Normal University, Nanchong, China. [3]School of Astronomy and Space Science, University of Chinese Academy of Sciences, Beijing, China. [4]Key Laboratory of Earth and Planetary Physics, Institute of Geology and Geophysics, Chinese Academy of Sciences, Beijing, China. [5]College of Earth and Planetary Sciences, University of Chinese Academy of Sciences, Beijing, China. [6]Qinghai Observing Station, Purple Mountain Observatory, Chinese Academy of Sciences, Delingha, China. ✉e-mail: licai@nao.cas.cn; fyang@nao.cas.cn; chenxiaodian@nao.cas.cn; hefei@mail.iggcas.ac.cn

**Table 1 | Comparison of key site characteristics with other known best sites in the world**

| Site | Median seeing (arcsec) | Air stability, ΔT 10–90% (°C) | Clear fraction (%) | Sky brightness (mag arcsec⁻²) | PWV <2 mm (%) |
|---|---|---|---|---|---|
| Lenghu | 0.75 | 2.7 | 70 | 22.0 | 55 |
| Mauna Kea | 0.75 | 6.8 | 76 | 21.9 | 54 |
| Cerro Paranal | 0.80 | 3.6 | 71 | 21.6 | 36 |
| La Palma | 0.76 | – | 84 | 21.9 | 21 |

Median seeing at Mauna Kea is from table 2 of ref. [1]. Median seeing at Cerro Paranal (1989–1995 and 1998–2002) and La Palma (1994–1995) are from table 1 of ref. [17]. The night temperature variations at Mauna Kea and Cerro Paranal are from ref. [1] and ref. [29], respectively. Here, ΔT 10–90% denotes the difference between the 90th and 10th percentiles of the temperature distributions. The temperature data for La Palma are not available. The cloud-free fractions of time (photometric time) at Mauna Kea, Cerro Paranal and La Palma are from table 2 of ref. [1], ref. [29], and table 4 of ref. [30], respectively. The sky brightness at Mauna Kea, Cerro Paranal and La Palma are from table 2 of ref. [31]. The fractions of PWV <2 mm at Mauna Kea, Cerro Paranal and La Palma are from table 2 of ref. [1], ref. [29], and table 1 of ref. [23], respectively.

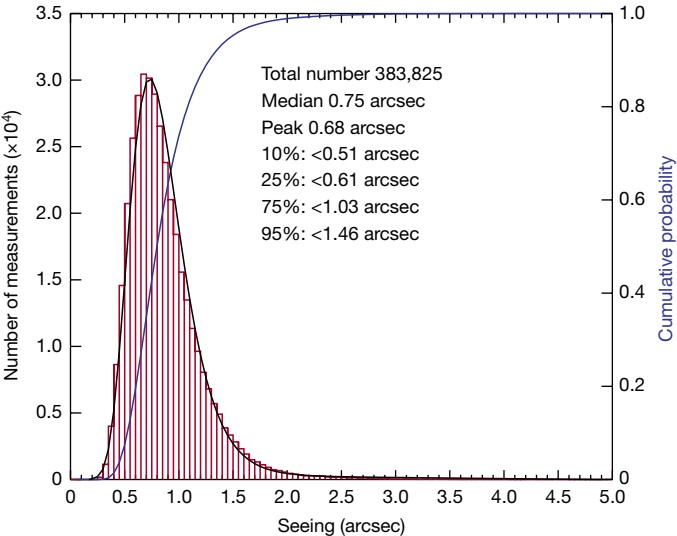

Total number 383,825
Median 0.75 arcsec
Peak 0.68 arcsec
10%: <0.51 arcsec
25%: <0.61 arcsec
75%: <1.03 arcsec
95%: <1.46 arcsec

**Fig. 1 | The night seeing at the Lenghu site.** The DIMM seeing data are collected from October 2018 to December 2020. The histogram is in red and the cumulative probability is in blue. The black solid line fits the histogram with a log-normal distribution.

To evaluate the observable time at Lenghu, we used a homemade all-sky camera (LH-Cam) with a 12-mm fish-eye lens customized for this site[13]. All-sky images have been captured every 20 min during the day and every 5 min between dusk and dawn without interruption since March 2018, regardless of the weather conditions. Another measure to evaluate the observable time makes use of the same SQM[14]. The SQM reading changes smoothly with the rotation of the starry sky during a clear night, and any cloud passage through the visible sky modifies the sequence of SQM magnitudes, resulting in a chaotic light curve (Methods, Extended Data Fig. 2). Our SQM photometer thereby enables us to study the overall cloudiness with a 1-min cadence. Combining LH-Cam and SQM data, we were able to reliably measure the clear time at the site. Observational data from 2018 to 2020 show that the site can provide, on average, over 90 fully clear photometric nights per year, 240 nights per year with more than 4 h of contiguous fully clear time and more than 280 nights with at least 2 h of contiguous clear time of photometric conditions. The fractional photometric time at the Lenghu site is 70%, which is slightly lower than the other three sites in Table 1.

## DIMM seeing statistics

Seeing, the blurring of stars due to atmospheric turbulence along the light path, is one of the key parameters to assess the quality of observations at a site for seeing-limited scientific goals, and a good integrated seeing measurement using a DIMM is the starting point for advance observational modes applying adaptive optics[15]. Measurements by DIMM are widely used and have become a standard assessment for integrated atmospheric optical turbulence[7,8,16]. A multi-aperture scintillation sensor can provide further information on how the total seeing is composed of different layers in the atmosphere above the site[17]. The same make of DIMM (the one provided by Alcor-System) applied in this study has also been used for many different site-selection campaigns[18,19]. Measurements are calculated for an average wavelength of 550 nm and corrected to the zenith (airmass unity).

Figure 1 shows the histogram of seeing taken from October 2018 until the end of 2020. We collected nearly a half a million data points in total. After removing the data not taken at clear time (mostly with cirrus passage over the local sky), or with spurs due to hot spots on the detector (Methods), we have 383,825 valid measurements. The seeing measurements follow a log-normal distribution. The median seeing is 0.75 arcseconds, and about 75% of the data are below 1 arcsecond. The DIMM dome is operated remotely and is only allowed to open when the wind speed is 10 m s⁻¹ or lower, and the seeing measurement stops immediately when the wind speed exceeds 12 m s⁻¹ for 3 min or 15 m s⁻¹

for 1 min during observations. When the power or Internet connection was poor, and, of course, when weather turned bad, no DIMM data were taken. Seeing data were collected for 457 nights, evenly distributed during the whole period until 31 December 2020. We tested the temporal variation[20] and the wind dependence of seeing and found that the seeing is stable for most of the observable time. The prevailing wind direction at the site is around 280° throughout the year, which is also the wind direction for the best median seeing. The median seeing is below 0.7 arcseconds between wind directions of 255° and 324° (Extended Data Fig. 3).

For the extreme observational requirements of very faint and/or high-redshifted targets in cutting-edge astrophysical problems, both high spatial resolution and long-time exposure are needed. Good natural seeing is a critical requirement for adaptive optics to work, especially for large-aperture telescopes[15]. A comparison of the total seeing at Lenghu with other known best sites in the world is shown in Table 1. The median value at Lenghu is the same as that at Mauna Kea (0.75 arcseconds) and is better than those at Cerro Paranal and La Palma.

In terms of total seeing, Lenghu is comparable to the best-established sites (in Chile, Hawaii and the Canary Islands), and is clearly the best one on the Tibetan Plateau (Methods, Extended Data Table 2, Extended Data Fig. 4). The best sites, with the addition of Lenghu, form a network of the best-possible conditions in both seeing and observing duty cycle of the time domain, second only to conditions in Antarctica[16,17].

The length of the observable time that is good in terms of both seeing and clear time is a key parameter to determine when considering a potential future observing site. To quantitatively evaluate the quality of the Lenghu site, a site quality matrix is defined based on the length of contiguous available observing time (cAOT) and the median value of DIMM seeing (DMn) on each night (Methods, Extended Data Table 3). For Lenghu, this score is 65%, which is comparable to 66% for both Cerro Paranal (1999–2012) and La Silla (2000–2008).

## Air stability

Local air stability analysis based on our weather data has been carried out. The intranight temperature variation is one of the essential elements that make up the total seeing. As of the end of December 2020, we have collected weather data for 756 days at a temporal

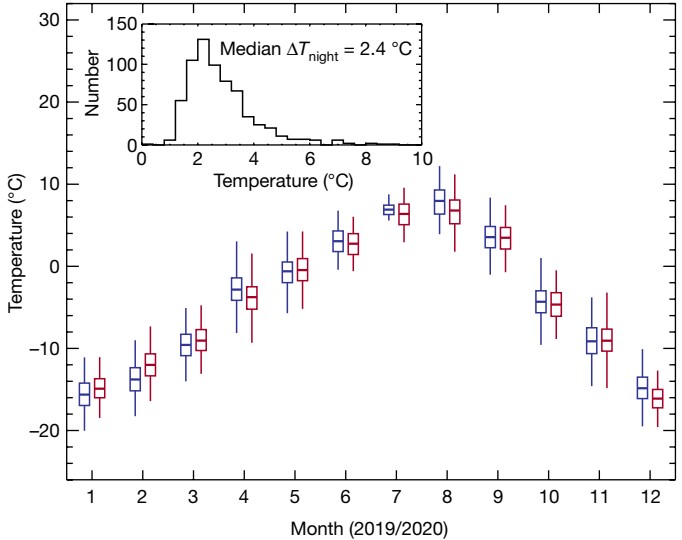

**Fig. 2 | Annual temperature variation pattern in 2019–2020.** The upper tip, upper top of the box, mid-bar in the box, bottom of the box and lower tip represent the standard deviation of the maximum temperature, the mean maximum temperature, the mean temperature, the mean minimum temperature and the standard deviation of the minimum temperature, respectively, on night for each month with blue for 2019 and red for 2020. The inset shows the histogram of the amplitude of the night temperature variations.

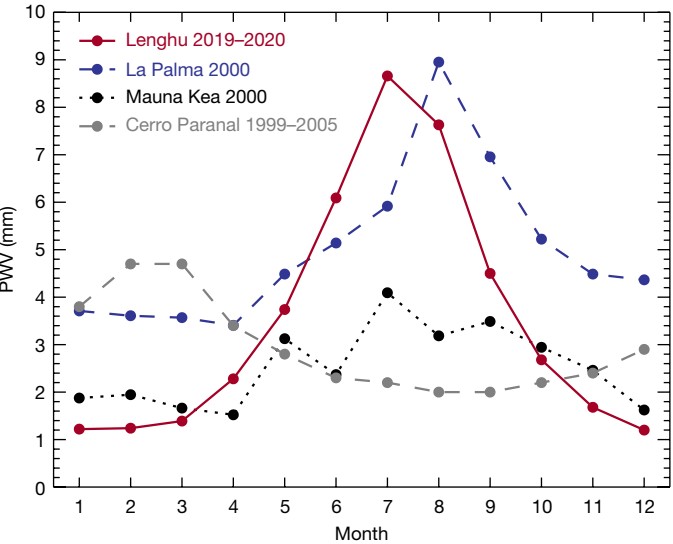

**Fig. 3 | Monthly averaged PWV at the Lenghu site in 2019–2020.** The reference curves of Mauna Kea (dotted) and La Palma (dashed) are taken from García-Lorenzo & Eff-Darwich[28]. The PWV of Cerro Paranal is taken from the ESO website. Dots are all monthly averaged values.

resolution of 1 min. Wind and temperature, and the spatial/temporal variabilities of both, affect local air stability dynamically and thermodynamically. The median wind speed at the site during the period is 4.5 m s$^{-1}$, indicating that the dynamic activity of the air at the site is rather low. The amplitudes of the temperature variation on an observing night directly reflect the air stability at the surface level of the site. For an amplitude of 10° or higher, the seeing is typically above 1.5 arcseconds, as we learnt from experience at a different site also on the Tibetan Plateau[13]. Figure 2 shows the annual temperature variation pattern with the intranight variation amplitude (peak to valley) indicated by vertical bars. In two full years, the average amplitude of the intraday temperature variation is only 5.6 K, and the average amplitude is only 2.4 K for the observing time between dusk and dawn. This is an advantage over all the sites surveyed for the Thirty Meter Telescope[1] and shown in Table 1. The median night temperature during winter (December, January and February) is −14.5 °C, which is comparable to the Ngari and Muztagh Ata test sites and much lower than the Daocheng test site[18]. Considering the mean warming trend of about 0.3 °C per decade at Lenghu[21], the median night temperature in winter will remain below −10 °C towards the end of this century.

## Turbulence profile

Optical turbulence along the light path from the top of the atmosphere to the site surface can be measured by several methods[22]. Limited by the power supply currently available at the site, direct measurements are yet to be routinely done. Traditional meteorological balloon experiments were carried out three times during the current stage of site selection. These missions measured atmospheric parameters that can be used to infer the turbulence strength along the path with acceptable accuracy. Balloon flights were performed in August and November 2020, representing typical conditions during summer and winter, which found that the height of the tropopause is around 11 km. Above 11 km, the refractive index structure constant $C_N^2$, where $N$ is the refractive index, decreases monotonically, meaning that the turbulence strength

decreases accordingly. The large fluctuations of the curves are due to the accuracy of the detectors. The $C_N^2$ is around $10^{-17.5}$–$10^{-17}$ m$^{-2/3}$ between 4 km and 11 km (Methods, Extended Data Fig. 5). On 16 November, the two turbulence profiles show a similar trend, but the turbulence strength at night (red profile) is lower than that during the dawn (grey profile). Between 6 km and 9 km altitude, the turbulence profile shows a clear difference in August and November, which suggests a possible seasonal pattern.

## Precipitable water vapour

All mega astronomical observing facilities designed now are aimed at cutting-edge scientific goals, such as the physics of the extremely early Universe and searching for signs of life on exoplanets. To realize such goals, ground-based observations are normally conducted through adaptive-optics-fed instruments working at infrared wavelengths. The PWV, the total amount of water vapour within the column between the telescope and the top of Earth's atmosphere, is a determining factor for such scientific goals.

At the Lenghu site, the infrastructure is still under construction; therefore, devices such as the scheduled infrared/submillimetre instruments cannot be supported on site yet. However, in both atmospheric studies and astronomical site qualifications[23,24], the PWV can be reliably modelled with an accuracy of better than 25% based on environmental quantities, including geographic and weather information (Methods). This method has been shown to be applicable for other sites on the Tibetan Plateau[4,25,26]. To anchor the modelled PWV to actual measurements, we applied Global Navigation Satellite Systems (GNSS) data[27] from a week in January 2021. This cross-check shows that our modelling of PWV using onsite weather station data is consistent with the GNSS method. Figure 3 shows the monthly mean PWV distribution modelled for the Lenghu site (Methods). The PWV is lower than 2 mm for 55% of the night. Compared with the PWV of the Mauna Kea and La Palma sites[28], Lenghu has a different pattern, possibly due to different climate patterns. During 2019–2020, the Lenghu PWV was high in summer and very low in winter. At night, PWV at the Lenghu site is better than that at Mauna Kea, meaning that Lenghu has a much higher potential than other known sites for infrared wavelengths (Table 1, Fig. 3).

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

## Methods

### Geographic information

The Lenghu site is at a local summit on Saishiteng Mountain, which is located to the east of the Altyn Mountains and on the northern edge of the Qaidam Basin. Its geographical coordinates are 38.6068° N, 93.8961° E, and it has an elevation of 4,200 m. The Lenghu site occupies a unique geographic position in the Eastern Hemisphere and bridges the huge gap between Mauna Kea (155.8246° W), Atacama (70.4042° W) and the Canary Islands (17.8577° W). This will form a perfect network of ground-based, high-quality observatories ready for great scientific discoveries, including searching signs of life on exoplanets, electromagnetic counterparts of gravitational wave outbursts, high-value transient events alerted by space-borne triggers that need to be done in very narrow time window[32,33] and much more.

According to the climate record collected at three local weather stations for past 30 years, the average annual precipitation is around 18 mm, with over 3,500 h of annual sunshine[34,35]. Land transport from the site to the local supporting base, Lenghu Town, and then to the developed areas of China by road and railway networks is convenient. The nearest international airport, highway access and the cargo railway stations of Dunhuang are all within only 250 km of the town. The altitude of Lenghu Town is only 2,700 m and is 80 km away from the site, which provides comfortable conditions for a supporting base for the site. This infrastructure enables good logistics for future activity at the site (Extended Data Fig. 1).

### AOT statistics

The clear night is derived by using both LH-Cam images analysis and the smoothness of integrated sky brightness records of the SQM. Lenghu has almost no artificial light pollution. Therefore, on LH-Cam images, any cloud will block the star background on a new moon night or when the moon is 18° or more under the horizon, leaving a dark patch on the image. When the moon is in the view of the camera, clouds will be directly visible. On sky brightness curves of the SQM, clouds make darkening or brightening fluctuations patterns depending on the moon age, and the amplitudes of such variations are correlated well with cloud coverage/thickness on the images[14], as demonstrated in Extended Data Fig. 2. In the left panel, three LH-cam images show a typical clear time (right), small clumpy cloud passage between clear time (left) and overcast (middle) cases, on the night of 6 October 2019, together with the SQM light curve. The right panel shows the distribution of clear time (cyan) in 2019 as an example of annual observing time statistics, based on the method described above.

### A comparison of Tibetan sites

In addition to Lenghu, other sites, namely Ngari, Muztagh Ata and Daocheng, on the Tibetan Plateau were also tested during an earlier general site survey by the Large Optical/infrared Telescope team. An intensive site testing programme was carried out for the Chinese 12-m telescope from 2016 to 2018 at the three sites. The testing results are concluded in an overview paper[18]. The Ngari site was later found to be good for the primordial gravitational wave project[4]. The concerns regarding strong wind, cloud cover in summer and light pollution from the nearby Shiquanhe Town[36] are potential challenges to further development of optical/infrared astronomy at the Ngari site. The other two sites are now being developed for different purposes. It turns out that Lenghu has the best observing conditions of all the sites tested on the plateau. A direct comparison of the key parameters of AOT and seeing are shown in Extended Data Table 2 and Extended Data Fig. 4, respectively. As the seeing data offered by the Large Optical/infrared Telescope team for Ngari, Muztagh Ata and Daocheng were all truncated at 3.0 arcseconds, the seeing data of Lenghu are processed accordingly, as presented in Extended Data Fig. 4.

In Extended Data Table 2, we adopted the method of the Large Optical/infrared Telescope team for AOT calculation, based on all-sky camera images. They divide the total visible sky by two circles with zenith angles of 44.7° and 65°, namely the inner and the outer circles. When there is no cloud in the inner and the outer circles, it is defined as 'clear' (or photometric in ref. [18]); when only the inner circle is clear, it is defined as 'outer' (spectroscopic).

### Site quality matrix scores

Extended Data Table 3 shows the site quality matrix for the Lenghu site based on all the nights that have both DMn and cAOT statistics. The DMn and cAOT are divided into five levels and four levels, respectively. Each element is assigned a weight according to the values of the DMn and cAOT (in parenthesis). The total score of the site is denoted by the ratio between the weighted summation of the number of nights and the total number of nights (457 in our case). For Lenghu, this score is 65%. An ideal site with all nights in yellow would score 100%. For the Delingha site, the median of all seeing measurements is 1.58 arcseconds[13]. Half of the AOT for Delingha (about 250 in total[13]) would be in blue (score 0.5) with the other half in brown (score 0.3), and the total score would be approximately 40%, which is typical for current existing classical observatories in China. For the Xinglong site (150 km from Beijing), where the Large Sky Area Multi-Object Fibre Spectroscopic Telescope (LAMOST) is hosted, the expected score would be even lower than the Delingha site for the same level of seeing, but with less observable time.

Using the publicly available seeing and AOT data from the European Southern Observatory (ESO) website, the sites at Cerro Paranal and La Silla are evaluated on the site quality matrix scale. Both the seeing and AOT data released on the ESO website are monthly average values. For the AOT of the ESO sites, a photometric night calls for six or more hours of consecutive photometric night. Therefore, only the third column of Table 1 (cAOT > 6 h) is used for the site quality estimations for the Cerro Paranal and La Silla sites. The monthly fractions of photometric nights are transferred to cAOT nights, and then the nights are divided into five levels according to the seeing divisions in Table 1. Finally, the total scores for Cerro Paranal (1999–2012) and La Silla (2000–2008) are both 66%. It is noted that these scores are upper limits for the sites as the nights with cAOT < 6 h (therefore, lower scores) cannot be assessed owing to the lack of daily weather data.

### Turbulence profiles

To understand the local meteorological pattern at the mountain region where our site is located, three balloon experiments were conducted at the Lenghu weather station. Once at 23:15 UT on 12 August 2020, and twice on 16 November 2020, at 11:18 UT and 23:44 UT. These balloon missions provided a vertical spatial resolution of 6.4 m. The mean potential temperature profile $\theta(h)$ is calculated by

$$\theta(h) = T(h)\left(\frac{P(h)}{1,000}\right)^{-0.286} \tag{1}$$

where $h$ is the altitude, $T(h)$ is the temperature profile in K and $P(h)$ is the pressure profile in hPa. The structure function of the temperature fluctuation $C_T^2$ is evaluated by the AXP model[37]. The refractive index structure constant $C_N^2$ is then estimated by the Gladstone formula

$$C_N^2(h) = C_T^2(h)\left(\frac{79 \times 10^{-6} P(h)}{T(h)^2}\right)^2 \tag{2}$$

The turbulence profiles calculated using the parameters obtained during the balloon flights are shown in Extended Data Fig. 5. Above 11 km, $C_N^2$ decreases monotonously with no seasonal pattern. $C_N^2$ is around $10^{-17.5}$ and $10^{-17}$ between 4 km and 11 km. On 16 November, the two turbulence profiles show a similar trend, but the turbulence strength at night (red profile) is lower than that in the morning (grey

profile). At an altitude of 6–9 km, the turbulence profile shows a clear difference in August and November, which suggests possible seasonal changes.

## PWV

The PWV can be calculated by the equation

$$\text{PWV} = \frac{1}{\rho g} \int_0^{p_z} q \, \mathrm{d}p \tag{3}$$

where $\rho$ is the density of liquid water, $g$ is the acceleration of gravity, $p_z$ is the pressure of the ground and $q$ is the specific humidity. The value of $q$ is calculated by the water vapour pressure $e$ by the equation

$$q = \frac{0.622e}{p - 0.378e} \tag{4}$$

The saturation water vapour pressure is usually converted from temperature by the Goff–Gratch formula[38]. We used the temperature, pressure and humidity of the ground weather station to estimate the amount of PWV. Here we adopted a temperature drop rate of 6.5 K km$^{-1}$, an exponential decay of air pressure with temperature, and the height of the tropopause is 11 km as measured by the balloon experiments (Extended data Fig. 5). The mean and median values of PWV modelled for the whole testing period are 3.13 mm and 2.01 mm, respectively (Extended Data Fig. 6).

By checking the data, we found that PWV changes substantially with season. We calculated the mean PWV by month and compared it with the PWV of La Palma and Mauna Kea (Fig. 3). Our two-year PWV values show a similar trend, that is, PWV values in winter are much lower than those in summer. The standard deviation in each month is about half of the average monthly PWV. From October to March, the mean PWV value is 1.55, which is 27% and 73% of the PWV values in La Palma and Mauna Kea, respectively[28].

To explore the possible deviations of our PWV, we also adopted the empirical equation between PWV and specific pressure of water vapour, $\text{PWV} = a_0 e + a_1$. The coefficients $a_0$ and $a_1$ change with elevation and latitude. Here we adopted the coefficients of Tibetan Plateau[25] (assuming an elevation of 4,200 m) and Ngari (also called Ali) site[26] (southwest part of the Tibetan Plateau) to re-estimate the PWV. Comparing the PWV estimates using the two sets of coefficients, our modelling of the PWV of the Lenghu site is consistent, but slightly overestimated by 0.15 mm and 0.01 mm, respectively.

## Dust grains

Dust and aerosol above an observing site can create problematic extinction for astronomical observations, and their presence in the ground layer can be troublesome for both optical surfaces and mechanical bearings. To measure local dust and aerosol, we implemented a dust meter (GRIMM EM180) in December 2019. We have so far collected a full year of uninterrupted data regarding the dust grains and aerosols of the site, with a temporal resolution of 5 min. The mean and median values of particulate matter with a diameter smaller than 10 μm ($PM_{10}$) density are 20.7 μg m$^{-3}$ and 11.7 μg m$^{-3}$, respectively. The ambient dust level of the Lenghu site is comparable to the sites in Atacama[39]. Twice in 2020, a high value of $PM_{10}$ was recorded, of around 100 μg m$^{-3}$, during

sandstorms that originated from the Taklimakan and local Gobi deserts. Dust grain densities higher than 50 μg m$^{-3}$ occurred 31 times in 2020, with a typical duration of several hours. Owing to the high altitude of the site, dust is less serious than at the La Palma site[40], which suffers from proximity to the Sahara Desert, and precautions can be implemented at the Lenghu site to protect the equipment for the few days per year affected by dust.

## Data availability

The seeing and weather data for the Lenghu site in 2018–2020 are available on a public website at http://lenghu.china-vo.org/index.html. The LH-Cam data are available from the corresponding authors on request. The monthly environmental parameters (seeing, cloudiness and PWV) for the Cerro Paranal and La Silla sites are publicly available from the ESO website at https://www.eso.org/gen-fac/pubs/astclim/paranal/. The tomographic data used in Extended Data Fig. 1 are provided by AW3D of the Japan Aerospace Exploration Agency (JAXA) available from https://www.eorc.jaxa.jp/ALOS/en/aw3d30/data/index.htm. (According to the data policy of JAXA, one can register a username and password to freely access the data.)

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

**Acknowledgements** This work is supported by the Lenghu municipal government. We acknowledge grants supporting this work from the Major Science and Technology Project of Qinghai Province 2019-ZJ-A10 and the National Natural Science Foundation of China (NSFC) 11633005. We thank the meteorological bureau of the local government for making balloon and historic weather data available. We are grateful to colleagues from the Purple Mountain Observatory of CAS for general discussions. Special gratitude is due to the organization committee of the Lenghu Industrial Park for constant support. We thank S. Justham for proofreading and suggestions.

**Author contributions** L.D. is the principal investigator of the project, who made the grand plan and participated in all instrumentation and data handling processes. F.Y. is the main person running the site monitoring system, and did most of the work on installation and maintenance, data taking and processing. X.C. did most of the modelling of PWV and the turbulence profile. F.H. did the GNSS data remote sensing of PWV and comparison. Q.L. performed on-site routine technical work. B.Z. did data archiving for this work. C.Z., K.W., N.L., A.R., Z.L. and J.P. took part in taking and reducing data. Z.Y. and J.T. worked on the initial hardware setup. J.P. also helped with the establishment of the site protection laws. All authors reviewed and commented on the manuscript.

**Competing interests** The authors declare no competing interests.

**Additional information**
**Correspondence and requests for materials** should be addressed to L.D., F.Y., X.C. or F.H.

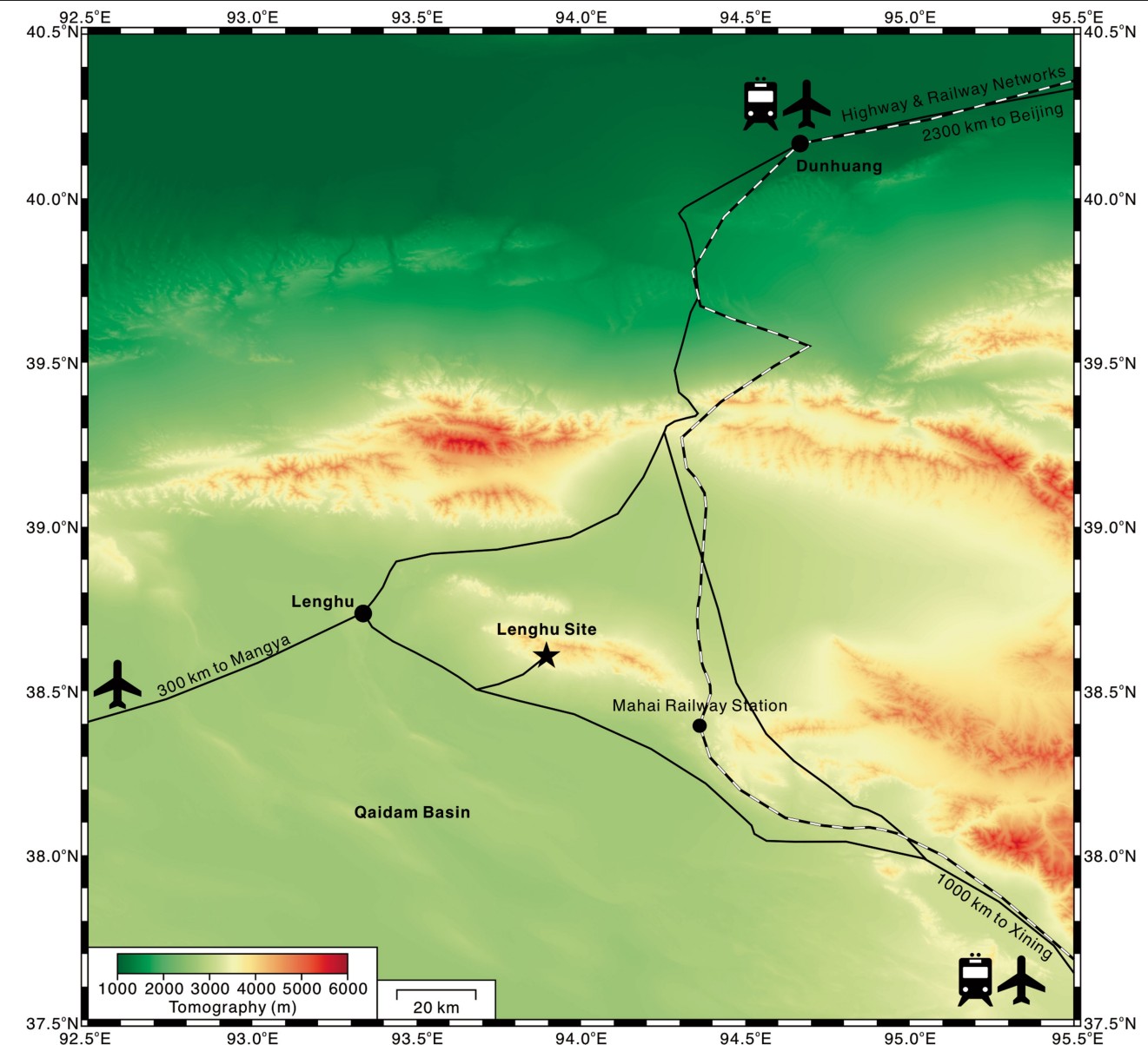

**Extended Data Fig. 1 | Geographic information for the Lenghu site.** The site is marked by the star. The national highways to Lenghu are shown by the black lines. The railway that connects Lenghu to the national railway network is shown by the dashed line. The nearest international airport in Dunhuang is 250 km away from Lenghu Town. The airport in Mangya is domestic.

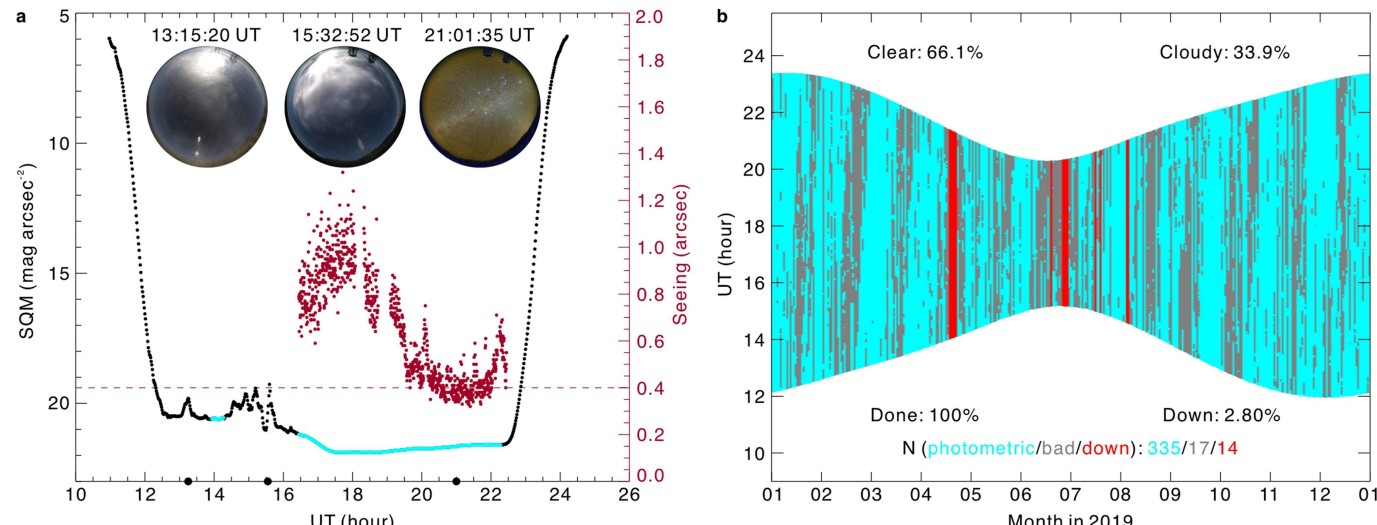

**Extended Data Fig. 2 | Sky background light curve and observable time statistics. a**, Definition of clear time. Three representative images of LH-Cam for 'clear', 'passage of cirrus' and 'cloudy' conditions are shown along with the light curve on the night on 6 October 2019. DIMM seeing data are also shown as red dots. **b**, Nightly distribution of clear (cyan), unclear (grey, cloudy and cirrus) and instrument down time (red) in 2019. Each vertical line represents the night between dusk and dawn in UT hours of one night.

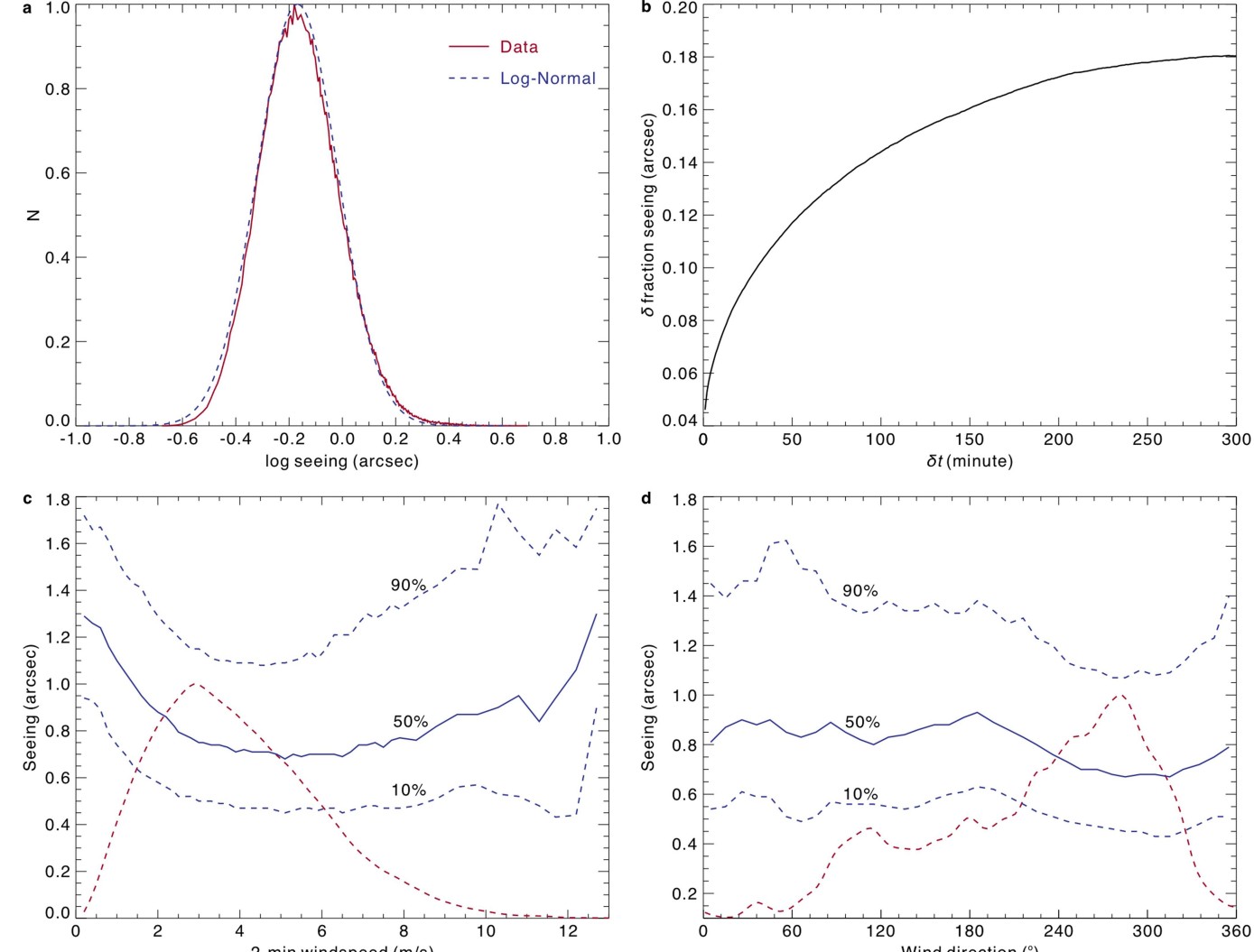

**Extended Data Fig. 3 | Temporal variation and the wind dependence of seeing. a**, Log-normal distribution of seeing. The maximum number (*N*) is normalized to 1. **b**, The fractional seeing variation between 1 min and 300 min according to the method of Racine[20]. **c**, Seeing versus 2-min wind speed.

**d**, Seeing versus wind direction. In **c**, **d**, the solid blue lines are the median and the two blue dotted lines are the 10% and 90% percentiles. The red dashed lines are the number distributions of wind speed and direction when we obtained the seeing measurements. The maximum number is also normalized to 1.

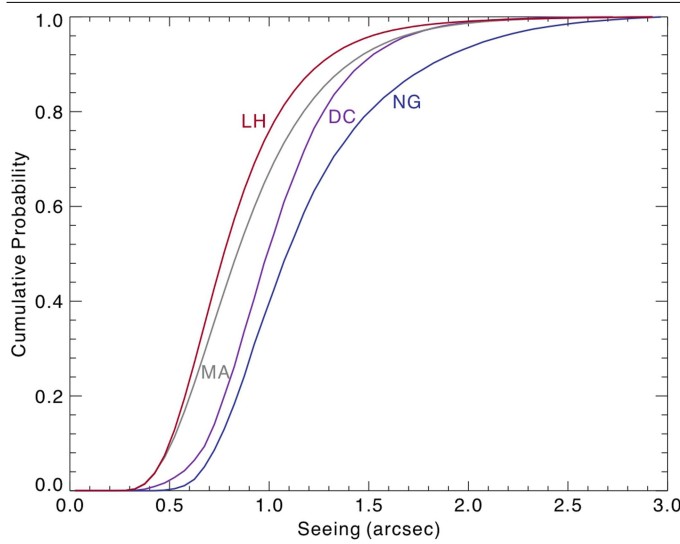

**Extended Data Fig. 4 | Comparison of the cumulative probability of seeing.**
The red, grey, purple and blue curves represent the distributions at Lenghu
(LH), Muztagh Ata (MA), Daocheng (DC) and Ngari (NG), respectively. As the
seeing data at MA, DC and NG are truncated at 3.0 arcseconds, the seeing data
at Lenghu are also truncated at 3.0 arcseconds for uniformity of comparison.

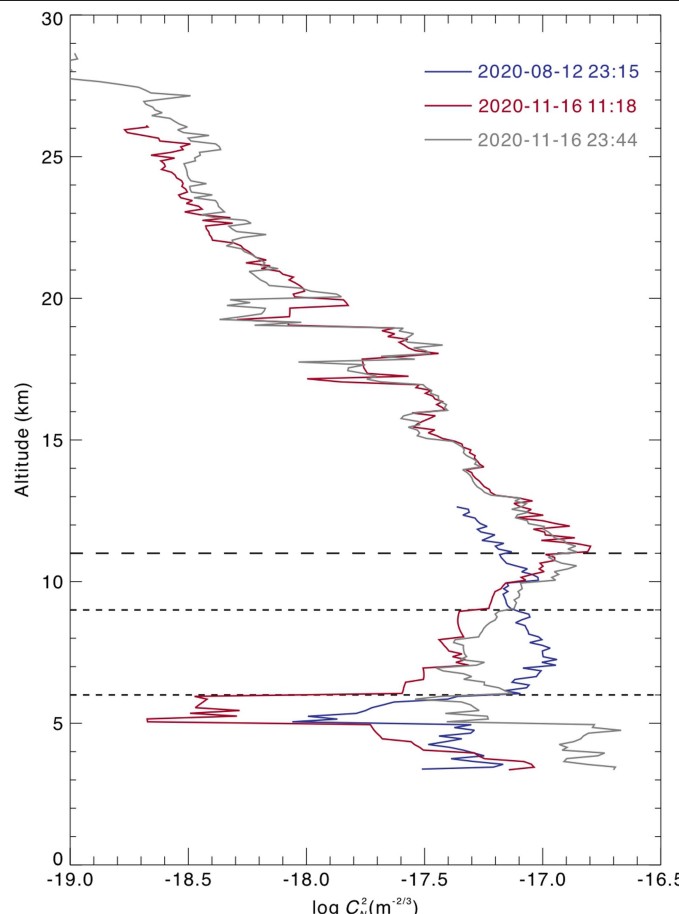

**Extended Data Fig. 5 | Turbulence profiles derived from balloon missions.**
The blue, red and grey profiles denote vertical distributions of the refractive
index structure constant $C_N^2$ at 23:15 UT on 12 August 2020 and 11:18 UT and
23:44 UT on 16 November 2020, respectively. The dashed line shows the height
of the tropopause at 11 km. The dotted lines denote heights of 6 km and 9 km.

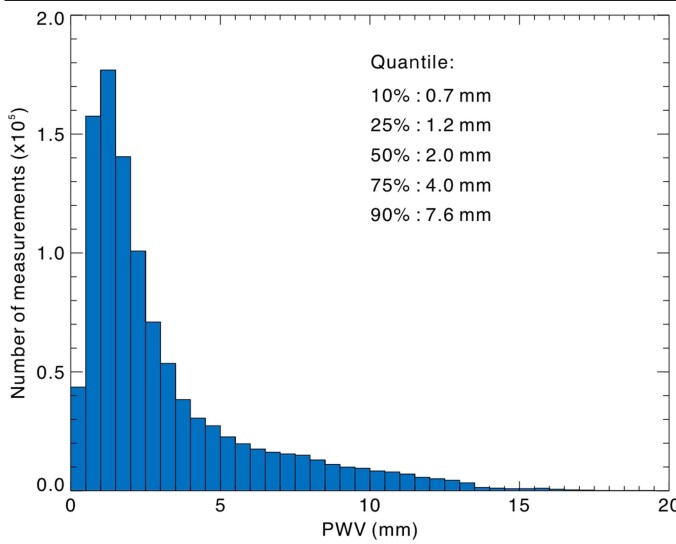

**Extended Data Fig. 6 | Histograms of PWV of the site.** Quantiles of the distribution are indicated.

**Extended Data Table 1 | List of start times of the site parameter measurements**

| Instrument* | Starting Date | Data Rate (min) | Derived Parameter* |
|---|---|---|---|
| Weather station | 22 September 2018 | 1 | Weather, PWV |
| SQM | 22 September 2018 | 1 | NSB, AOT |
| LH-Cam | 30 March 2018 | 20 (day)/5 (night) | AOT |
| DIMM | 28 October 2018 | 1 | Seeing |
| Dust Gain Counter | 28 December 2019 | 5 | Dust |
| Balloon Flights† | August-November 2020 | 3 flights | PWV, TP |

*Abbreviations: SQM–Sky Quality Meter, LH-Cam–Lenghu all sky Camera, DIMM–Differential image motion monitor, PWV–Precipitable Water Vapour; NSB–Night Sky Background brightness, AOT–Available Observing Time, TP–Turbulence Profile.

† Three balloon flights were carried out on 12 August 2020 (one flight) and 16 November 2020 (two flights), respectively. See Methods for detailed derivation of the parameters.

*SQM, sky quality meter; LH-Cam, Lenghu all-sky camera; DIMM, differential image motion monitor; PWV, precipitable water vapour; NSB, night sky background brightness; AOT, available observing time; TP, turbulence profile. †Three balloon flights were carried out on 12 August 2020 (one flight) and 16 November 2020 (two flights), respectively. See Methods for detailed derivation of the parameters.

**Extended Data Table 2 | Comparison of key parameters between Tibetan sites**

| Site | cAOT*>3 h | AOT*>3 h | 25% (arcsec) | 50% (arcsec) | 75% (arcsec) |
|---|---|---|---|---|---|
| Lenghu | 73% | 84% | 0.61 | 0.75 | 1.03 |
| Ngari (Ali) | 72% | 82% | 0.88 | 1.08 | 1.39 |
| Muztagh Ata | 63% | 73% | 0.64 | 0.84 | 1.10 |
| Daocheng | 52% | 59% | 0.82 | 0.99 | 1.21 |

*AOT is the total observing time in a night (photometric and spectroscopic, as in ref. [18]), cAOT is contiguous AOT in a night. The percentages are number of nights fulfill the conditions divided by 365.

*AOT is the total observing time in a night (photometric and spectroscopic, as in ref. [18]) and cAOT is contiguous AOT in a night. The percentages are the number of nights that fulfil the conditions divided by 365.

**Extended Data Table 3 | Site quality matrix for Lenghu**

| DMn (arcsec) | AOT (nights) | cAOT>6 h | cAOT 4-6 h | cAOT 2-4 h | cAOT<2 h |
|---|---|---|---|---|---|
| <0.60 | 61 | 26 (1) | 10 (0.8) | 12 (0.7) | 13 (0.6) |
| 0.60−0.80 | 174 | 87 (0.8) | 24 (0.7) | 42 (0.6) | 21 (0.5) |
| 0.80−1.00 | 140 | 67 (0.7) | 39 (0.6) | 16 (0.5) | 18 (0.4) |
| 1.00−1.50 | 70 | 36 (0.6) | 11 (0.5) | 17 (0.4) | 6 (0.3) |
| >1.50 | 12 | 5 (0.5) | 2 (0.4) | 5 (0.3) | 0 (0.1) |

The numbers in the brackets denote the site quality scores and the different colours are coded to distinguish different levels of site quality scores.