## [Peer Review File · Nature]

Manuscript Title: Lenghu on the Tibetan Plateau as an astronomical observing site

Reviewer Comments & Author Rebuttals

Reviewer Reports on the Initial Version:

Referees' comments:

Referee #1 (Remarks to the Author):

A. Key results: Please summarise what you consider to be the outstanding features of the work.

This article describes the results of three years of site-testing of a potential location (Lenghu) in China for a future large optical/infrared telescope. The authors should be congratulated on completing this difficult task at such a remote location. Their work in turn builds on some two decades of site-testing in China, by many dozens of people.

The choice of an observatory location is critically important since it has a large influence on the quality of the science that can be done, and it is not something that can be altered once construction starts.

The site in China is particularly important for astronomy given that China is likely to invest a large amount of money there (a good fraction of a billion dollars), and the location of the site fills an important gap in the worldwide time coverage of astronomical events (i.e., if a transient event occurs, having coverage at this longitude is very valuable). The article provides convincing evidence that the Lenghu site is indeed comparable in quality to the best established sites (such as Chile, Hawaii, and the Canary Islands) for large telescopes.

B1. Validity: Does the manuscript have flaws which should prohibit its publication? If so, please provide details.

No. The paper is well written and comprehensive.

B2. Originality and significance: If the conclusions are not original, please provide relevant references. On a more subjective note, do you feel that the results presented are of immediate interest to many people in your own discipline, and/or to people from several disciplines?

The results are original, and will be of considerable interest to many astronomers. The information is timely, since site selection for China's Large Optical Telescope is presumably very close to being finalised.

One issue that worries me is the overall site testing situation in the Tibetan plateau region.. Line 38 mentions that "Previous surveys ... have identified candidate sites..." but ref 7 really only talks about the Ngari site (also called Ali), so that is just a single site, not "sites". Are there any other sites under consideration in Tibet? I also note that (as far as I can see) there are no authors in common between this article and ref 24 "Site testing campaign for the Large Optical/infrared Telescope of China: Overview" - which makes me wonder about the politics of the situation of site-testing in China. It would help to know if there are really just two sites in Tibet being considered (Ngari and Lenghu), or are there others?

C. Data & methodology: Please comment on the validity of the approach, quality of the data and

quality of presentation. Please note that we expect our reviewers to review all data, including any extended data and supplementary information. Is the reporting of data and methodology sufficiently detailed and transparent to enable reproducing the results?

The data and methodology of the paper seem perfectly fine. There are a dozen or so typos and other editorial changes that I haven't bothered to mention since the Nature editors will find them quickly. I list below some comments, with line numbers.

Line 30, the article compares Lenghu with the "best sites known for astronomy across the world", but does not mention Antarctica, which is well-established now as containing considerably better sites for most types of astronomy. In particular, China itself has Kunlun Station at Dome A, which is at the best location within Antarctica, and the subject of a recent Nature article (Ma, et al. 2019, Nature 583, 771-774). This paper is in the reference list, but not discussed in any detail. The authors should acknowledge more fully the existence of Dome A at least, and the superior site properties there. The reason that Dome A is not a suitable choice for China's large telescope is simply that the logistical challenges of building it there are too great, without a pathfinder smaller-scale telescope such as the proposed KDUST telescope.

Line 46, yes, the unique geographic position of Lenghu is certainly a big advantage. It would be worth mentioning that Dome A has even better time-coverage than Lenghu, due to its location close to the South Pole, although there is an obvious trade-off between the length of time that you can observe a given position on the sky, and the amount of sky coverage that you have.

Line 54, "huge area" should be quantified.

Line 103, the SQM does not accurately measure light from all angles above the horizon as the article states. It is quite peaked towards the zenith, and it has less than 20% of its peak responsivity at zenith distances greater than 60 degrees. The manufacturer says that the effective solid angle is about 1.53 rads, which is only 24% of a hemisphere.

Line 132 implies that ref 23 (Lawrence et al) used a DIMM. This reference in fact used a MASS and a sonic radar, not a DIMM.

Line 132 should specify the exact model of DIMM used; ref 24 talks about three different models, and ref 25 describes a range of instruments, so it is not possible to determine unambiguously the model used.

Line 134, really 550nm? 500nm is used in ref 24.

Lines 155, 162, 163, 260 the observatory scores should be given to 2 sig figs. The 3rd digit is not justified (i.e., annual variations, and some details of the comparing the data from the sites, will certainly cause it to fluctuate).

Line 305, the data are supposed to be available to the public here: <https://lenghu.china-vo.org/index.html> but this website gives a security warning, and when bypassed, it then gives a "404 Error - Page Not Found". This should be fixed prior to publication.

Line 310, the Japanese data site requires a username/password; perhaps indicate how this can be obtained.

Line 409, Fig 1, with 390K points takes a long time to plot, which makes scrolling around the article tedious. Also, artifacts are present depending on the program used and the zoom setting (e.g., Word displays spurious stripes at some zoom settings). To fix this I suggest using an alternative way of displaying the data. More importantly, it would be helpful to consider replacing this figure with a cumulative plot, such as the one in Lawrence et al. 2004, and fig 1 in Ma et al

2019. A cumulative plot allows the simple extraction of important information, such as "what fraction of the time is the seeing better than 0.5 arcseconds?". Given the critical importance of the fraction of time when the seeing is superb, a cumulative plot is most informative. The plot should show for comparison the Dome A values, and those for Chile, Hawaii, etc.

Line 426, just a comment that the annual temperature variation at Lenghu is remarkably low, particularly the very low range during a day/night cycle.

Line 426, in fig 2, consider separating the data for 2019 and 2020 to make the comparison clearer.

Line 458, why show lines at 6 and 9 km?

D. Appropriate use of statistics and treatment of uncertainties: All error bars should be defined in the corresponding figure legends; please comment if that's not the case. Please include in your report a specific comment on the appropriateness of any statistical tests, and the accuracy of the description of any error bars and probability values.

Yes, all fine, apart from the extraneous sig figs mentioned above.

E. Conclusions: Do you find that the conclusions and data interpretation are robust, valid and reliable?

Yes.

F. Suggested improvements: Please list additional experiments or data that could help strengthening the work in a revision.

Addressed above.

G. References: Does this manuscript reference previous literature appropriately? If not, what references should be included or excluded?

Yes.

H. Clarity and context: Is the abstract clear, accessible? Are abstract, introduction and conclusions appropriate?

Yes to all.

Referee: Michael Ashley, University of New South Wales

Referee #2 (Remarks to the Author):

This paper introduces Lenghu area as a potential reservoir of sites for ground based astronomy in the eastern part of the Tibetan plateau. One of the summit has been thoroughly tested in the past two years, an impressive database of several key parameters has been collected and some results of the analysis are presented. The authors aim at comparing the astroclimate properties of this site to those of a few major reference ground based observatories in operation. They show convincingly that it is in many aspects of comparable quality for future scientific projects. The text is clear and the conclusions are well documented, I consider nevertheless the paper is too modest and should be expanded in some domains to reach the level of expectancy of Nature readers, suggestions are detailed in what follows.

--Introduction

Many readers are not aware of astronomical site selection activities in China, putting the present survey in a global perspective would be helpful: the project should mention in addition to Ali, the other potential sites in the western Tibetan plateau which were part of the site search for the Chinese 12m project. It is obvious, looking at the attached map, that Lenghu had a unique geographical position, filling a gap in the area.

--Water Vapor

Ground based observations in the infrared suffer of the background noise created by the radiation of the telescope structure and components, in particular on "warm" sites like in the coastal Atacama. Hence the very low water vapor period can only be used if the telescope is cooled, either actively which is unrealistic for ELTs, or passively if the site is cold enough. It appears that Lenghu has an enormous advantage in this respect, being the coldest of the Tibetan sites in winter (Ali median is -10C, the 2 others -5C) precisely when the PWV is at the lowest.

Moreover, with global warming predictions of about .5C per decade (attach.2), some of the warmer sites could well reach positive winter temperature within this century.

--Seeing

The authors have collected an impressive amount of continuous DIMM (please correct page 3 line 82 DIMM = Differential Image Motion Monitor) data which should allow to extract much more than the median 1mn seeing: the median seeing expected to be achieved as a function of the exposure time is also relevant. As it is clear that the Lenghu seeing is variable with a long tail in the distribution, it is important to know if the spikes occur as groups or rather aleatory. An example of such an analysis is given by R. Racine in Mauna Kea in "Temporal fluctuations of atmospheric seeing" 1996, PASP 108, 372-374.

Secondly, it is important to be able to estimate which fraction of the seeing is close to the ground, hence removable on wide field with GLAO technique. The seeing rose, polar plot of the seeing records versus simultaneous incoming wind direction, can help. Should the bad seeing be always linked to the same wind direction, it is clearly caused by local features and thus located close to the surface.

Congratulation to the Lenghu monitoring team for this achievement

Author Rebuttals to Initial Comments:

Dear Reviewers,

We greatly appreciate your efforts in the previous versions of the manuscript. We have made point-to-point responses to all the comments/suggestions raised in your review reports and made the corresponding revisions in the context. All the replies in this document are colored by blue and the revisions/changes in the revised manuscript are marked in red.

Sincerely yours,

The authors.

General Response to Referees' comments:

To facilitate our response, a map provided by Referee #2 (we appreciate that very much!) is attached here to illustrate the three candidate sites for the Chinese 12m LOT project together with Lenghu. This map shows where exactly the sites are located, and the relative accessibilities among them in the context of logistics and support for future developments. The blue lines only connect the sites, to direct eyes. They are all state roads (mostly paved), except the Ngari-Tashkurgan part, which is a gravel road, and sometimes just truck tire prints in wilderness. Please also refer to the map in the extend data part for the transportation manners to reach Lenghu.

-----Responses to Reviewer #1-----

Referee #1 (Remarks to the Author):

A. Key results: Please summarise what you consider to be the outstanding features of the work.

This article describes the results of three years of site-testing of a potential location (Lenghu) in China for a future large optical/infrared telescope. The authors should be congratulated on completing this difficult task at such a remote location. Their work in turn builds on some two decades of site-testing in China, by many dozens of people.

The choice of an observatory location is critically important since it has a large influence on the quality of the science that can be done, and it is not something that can be altered once construction starts.

The site in China is particular important for astronomy given that China is likely to invest a large amount of money there (a good fraction of a billion dollars), and the location of the site fills an important gap in the worldwide time coverage of astronomical events (i.e., if a transient event occurs, having coverage at this longitude is very valuable). The article provides convincing evidence that the Lenghu site is indeed comparable in quality to the best established sites (such as Chile, Hawaii, and the Canary Islands) for large telescopes.

Response:

Thank you, Michael, for the comment.

B1. Validity: Does the manuscript have flaws which should prohibit its publication? If so, please provide details.

No. The paper is well written and comprehensive.

B2. Originality and significance: If the conclusions are not original, please provide relevant references. On a more subjective note, do you feel that the results presented are of immediate interest to many people in your own discipline, and/or to people from several disciplines? The results are original, and will be of considerable interest to many astronomers. The

information is timely, since site selection for China's Large Optical Telescope is presumably very close to being finalised.

One issue that worries me is the overall site testing situation in the Tibetan plateau region.. Line 38 mentions that "Previous surveys ... have identified candidate sites..." but ref 7 really only talks about the Ngari site (also called Ali), so that is just a single site, not "sites". Are there any other sites under consideration in Tibet? I also note that (as far as I can see) there are no authors in common between this article and ref 24 "Site testing campaign for the Large Optical/infrared Telescope of China: Overview" - which makes me wonder about the politics of the situation of site-testing in China. It would help to know if there are really just two sites in Tibet being considered (Ngari and Lenghu), or are there others?

Response:

Thank you for paying attention to site survey activities in West-China.

More words are needed to explain your concern, but first of all, thank you for pointing out this.

Since 2000, site surveys (restarted many times for complicated issues, one of them has been funding) have been carried out. The tested sites are: Ngari (Ali), Muztagh Ata, Daocheng (indicated on the map above), and other spots also on the plateau tested only for very short periods and then abandoned. These actions were all before 12m LOT proposal (2015). Then, the three sites were officially included in the grand plan of LOT in 2016. The site testing at Lenghu is a separate task, independent of LOT site project. The first author of this work (LD) is one of the 3 "external observers" appointed by LOT committee. In 2020 the site team published 10 papers (RAA Vol20 No6, the overview paper cited in the current work) as a mini volume. I was then invited to review the manuscripts. For this reason, there is no common names between those papers and the current one.

[Redacted]. One can never imagine (me too) that the Daocheng site was selected (by vote) for 12m LOT in the latest proposal (in 2019, when Lenghu is still being tested). As in the overview paper (ref24) of the mini volume in RAA, Daocheng is the worst one among the 3 sites even in common sense (unusable summer, and very humid winter).

Ref7 (now ref.8) is a Science journal report about the early site testing in Ngari region. From the report, the Japanese participants raised a few issues on light pollution and strong wind at the site. As of today, the local city brightened up a lot, and the subsequence testing work did

not go very smoothly. Later in 2018, an astro-particle team started a project for Primordial Gravitational Wave (PGW) project (new ref.7, also a Science journal report by Normile in 2019), claimed a super good site. However, that is not really “Astronomical Observatory”. As it turned out later, that both city light of Shiquanhe and strong winds are troubles. Many telescope projects originally planned for Ngari station were moved out from that site, some of them (telescopes of Nanjing University time-domain project, for instance) are recently relocated to Lenghu site.

Lenghu, as a future site, was never into the 12m LOT scope in the previous site selection campaign, the main reason was that Lenghu site monitoring started slight later (end of 2017) than the 3 sites. However, as of the end of 2020, Lenghu attracted a lot of attention from the decision-making level, mostly due to its extraordinary results in comparison to the 3 sites of LOT, and in addition the excellent accessibility. In the latest development (after this paper’s submission), A Magellan type 6.5m telescope is being considered to come to Lenghu; LAMOST (currently in Xinglong) will be upgraded to 8m aperture/15k fibers, and then will be relocated also to Lenghu. Although 12m LOT proposal is still waiting for final funding, but once it will be decided, I believe it will also come to Lenghu (already being discussed at NAOC and CAS, the institution responsible for building and operating LOT). This makes this work even more timely.

A table comparing Lenghu and the three sites in terms of seeing and observing time is now added to the extended data part, based on a processed data set provided by LOT site paper. Seeing measurements for all the 4 sites were taken using the same instruments and software. Observing time estimation are made on the same method of LOT in this comparison.

Revision in the manuscript:

(Lines 151-154) In terms of total seeing, Lenghu is comparable with best established sites (in Chile, Hawaii, and the Canary Islands), and is obviously the best one on the Tibetan Plateau (see Methods, Extended Data Table 2, and Extended Data Figure 4). On the surface of the earth, the best conditions in both seeing and observing duty cycle for time-domain is clearly on Antarctica^{5,6}.

We added a new section in Methods (Lines 266-278):

Comparison of Tibetan sites: key parameters

In addition to Lenghu, 3 other sites on different locations on the Tibetan Plateau were also tested for different purposes since 2000. An intensive site testing program was carried out for Chinese 12 m telescope recently at these 3 sites, namely Ngari, Muztagh Ata and Daocheng, and are concluded in the overview paper²⁴. It turns out that Lenghu has the best observing condition for optical/infrared. The direct comparisons of the key parameters of AOT and seeing are shown in Extended Data Table 2 and Extended Data Fig. 4, respectively. It is noted that, since the seeing data at Ngari, Muztagh Ata and Daocheng are all truncated at 3.0 arcseconds, the seeing data at Lenghu is also truncated at 3.0 arcseconds in Extended Data Fig. 4.

In Extended Data Table 2, the AOT is calculated using the LOT method based on all sky camera images. They divide the total visible sky by two circles with zenith angle 44.7° and 65° , namely the inner and the outer circle. When there is no cloud in the outer circle, it is defined as ‘clear’ (otherwise photometric in ref. 24); and it is ‘outer’ if only the inner circle is clear (spectroscopic).

Extended Data Table 2. Comparison of key parameters between Tibetan sites.

Site	cAOT*>3 h	AOT*>3 h	25% (arcsec)	50% (arcsec)	75% (arcsec)
Lenghu	73%	84%	0.61	0.75	1.03
Ngari (Ali)	72%	82%	0.88	1.08	1.39
Muztagh Ata	63%	73%	0.64	0.84	1.10
Daocheng	52%	59%	0.82	0.99	1.21

*AOT is the total observing time in a night (photometric and spectroscopic, as in ref. 24), cAOT is contiguous AOT in a night. The rates are the number of nights fulfill the conditions divided by 365.

Extended Data Fig. 4. | Comparison of the cumulative probability of seeing. The red, grey, purple, and blue curves represent the distributions at Lenghu (LH), Muztagh Ata (MA), Daocheng (DC), and Ngari (NG), respectively. Since the seeing data at MA, DC and NG are truncated at 3.0 arcseconds, the seeing data at Lenghu are also truncated at 3.0 arcseconds for uniformity of comparison.

C. Data & methodology: Please comment on the validity of the approach, quality of the data and quality of presentation. Please note that we expect our reviewers to review all data, including any extended data and supplementary information. Is the reporting of data and methodology sufficiently detailed and transparent to enable reproducing the results?

The data and methodology of the paper seem perfectly fine. There are a dozen or so typos and other editorial changes that I haven't bothered to mention since the Nature editors will find them quickly. I list below some comments, with line numbers.

Response:

Thanks. We already spotted the typos, all corrected.

Line 30, the article compares Lenghu with the "best sites known for astronomy across the world", but does not mention Antarctica, which is well-established now as containing considerably better sites for most types of astronomy. In particular, China itself has Kunlun Station at Dome A, which is at the best location within Antarctica, and the subject of a recent

Nature article (Ma, et al. 2019, Nature 583, 771-774). This paper is in the reference list, but not discussed in any detail. The authors should acknowledge more fully the existence of Dome A at least, and the superior site properties there. The reason that Dome A is not a suitable choice for China's large telescope is simply that the logistical challenges of building it there are too great, without a pathfinder smaller-scale telescope such as the proposed KDUST telescope.

Response:

Thank you for mentioning this, we have good connection with Zhaohui during this work. we highly valued the Dome A paper. We regarded Dome A as a quasi-space program, so that we did not even try to match the results. But as you suggested, a short description is added.

Revision in the manuscript:

(Lines 151-154) In terms of total seeing, Lenghu is comparable with best established sites (in Chile, Hawaii, and the Canary Islands), and is obviously the best one on the Tibetan Plateau (see Methods, Extended Data Table 2, and Extended Data Figure 4). On the surface of the earth, the best conditions in both seeing and observing duty cycle for time-domain is clearly on Antarctica^{5,6}.

Line 46, yes, the unique geographic position of Lenghu is certainly a big advantage. It would be worth mentioning that Dome A has even better time-coverage than Lenghu, due to its location close to the South Pole, although there is an obvious trade-off between the length of time that you can observe a given position on the sky, and the amount of sky coverage that you have.

Response:

Thank you. Regarding time-domain science in connection with Dome A and Dome C, some wording is included **(Lines 151-154)**.

Line 54, "huge area" should be quantified.

Response:

Thank the reviewer for the comment. We estimated the area with altitude below 3,000 m around Lenghu Town using the tomographic data presented in Extended Data Figure 1. The area is over 100 thousand square kilometers.

Revision in the manuscript:

(Lines 54-56) The huge area (**over 100,000 square kilometers**) surrounding Lenghu Town has a relatively low altitude below 3,000 m, the climate is extremely arid, like the Atacama Desert, and the local sky (day and night) has long been known to be unusually clear¹².

Line 103, the SQM does not accurately measure light from all angles above the horizon as the article states. It is quite peaked towards the zenith, and it has less than 20% of its peak responsivity at zenith distances greater than 60°. The manufacturer says that the effective solid angle is about 1.53 rads, which is only 24% of a hemisphere.

Response:

Yes, the model SQM-LE deployed at Lenghu follows the specifications of the manufacturer. It detects light from lower altitudes (zenith angle > 60) at very low efficiency. We changed wording in the text.

Revision in the manuscript:

(Lines 104-106) ...accurately measures the integrated light of the entire visible sky, with the sensitivity optimized toward the zenith and quickly drop-off to less than 20% once the zenith angle is greater than 60°.

Line 132 implies that ref 23 (Lawrence et al) used a DIMM. This reference in fact used a MASS and a sonic radar, not a DIMM.

Response:

Thank you for the comment. We have specified this in the manuscript. In our plan, a MASS (being built by our colleagues from NIAOT) was expected to operate no later than middle of 2020. Unfortunately, the pandemic slowed down quite a few things in our project including this one.

Revision in the manuscript:

(Lines 131-134) Measurements by DIMM are nowadays widely used and have become a standard assessment for integrated atmospheric optical turbulence^{5,16,17}. **Multi-aperture scintillation sensor (MASS) can provide further information on how the total seeing is composed of from different layers in the atmosphere above the site⁶.**

Line 132 should specify the exact model of DIMM used; ref 24 talks about three different models, and ref 25 describes a range of instruments, so it is not possible to determine unambiguously the model used.

Response:

Thank you. Ref. 24 applied 3 different types, one of them is referred as “French DIMM”. The other two are either manufactured by NIAOT or home-made by the Ngari team. Inter-calibrations were made between all the systems before and during the campaign. The “French DIMMs” used by LOT team and by us are all from Alcor-System which comes with the same optical/mechanical design and software, as in ref25. It is the only full-system model offered by the company.

Revision in the manuscript:

(Lines 134-136) The same make of DIMM **(the one provided by Alcor-System)** applied in this work is also used for many different site selection campaigns^{24,25}. Measurements are calculated for average wavelength 550 nm and corrected to the zenith (airmass unity).

Line 134, really 550nm? 500nm is used in ref 24.

Response:

The DIMM at Lenghu site operates at 550 nm. We have double checked with the manufacturer Alcor-System. The wavelength of 500 nm was adopted in ref. 24, I believe that is not correct.

Lines 155, 162, 163, 260 the observatory scores should be given to 2 sig figs. The 3rd digit is not justified (i.e., annual variations, and some details of the comparing the data from the sites, will certainly cause it to fluctuate).

Response:

You are right, thanks. All the values are given to 2 sig figs.

Line 305, the data are supposed to be available to the public here: <https://lenghu.china-vo.org/index.html> but this website gives a security warning, and when bypassed, it then gives a "404 Error - Page Not Found". This should be fixed prior to publication.

Response:

Thank you for pointing out this. For security reasons, our web administrator shutdown all ports but the classical http ports: 80 and 8080. Therefore, one should always use http instead of https. We have double-checked the data link by inviting several domestic and foreign persons to click the link. All of them can access the website normally.

As we presented in the Data Availability, the correct website is: <http://lenghu.china-vo.org/index.html>. NOT <https://lenghu.china-vo.org/index.html>.

In case the problem will be persistent, we can use other way to make our data publicly available.

Line 310, the Japanese data site requires a username/password; perhaps indicate how this can be obtained.

Response:

Thank you for pointing out this. According to data policy of JAXA, we cannot distribute the data directly. But one can easily register a username and password to freely download the data for academic use. We have added a sentence in the Data Availability to point out this.

Revision in the manuscript:

(Lines 341-341) The tomographic data used in Extended Data Figure 1 are provided by AW3D of the Japan Aerospace Exploration Agency (JAXA) available from <https://www.eorc.jaxa.jp/ALOS/en/aw3d30/data/index.htm> (According to the data policy of JAXA, one can have free access simply by registering a username and password.)

Line 409, Fig 1, with 390K points takes a long time to plot, which makes scrolling around the article tedious. Also, artifacts are present depending on the program used and the zoom setting (e.g., Word displays spurious stripes at some zoom settings). To fix this I suggest using an alternative way of displaying the data. More importantly, it would be helpful to consider replacing this figure with a cumulative plot, such as the one in Lawrence et al. 2004, and fig 1 in Ma et al 2019. A cumulative plot allows the simple extraction of important information, such as "what fraction of the time is the seeing better than 0.5 arcseconds?". Given the critical importance of the fraction of time when the seeing is superb, a cumulative plot is most informative. The plot should show for comparison the Dome A values, and those for Chile, Hawaii, etc.

Response:

We greatly appreciate your suggestion. Fig. 1 has been revised to display the histogram and the cumulative curve. The fraction of time that the seeing is better than 0.51, 0.61, 1.03, and 1.46 arcseconds are also shown in the figure. These values are comparable to the current Mauna Kea and Chile web broadcasting values. The best seeing is surely not so good as Dome A.

Revision in the manuscript:

Figure 1 | The night-time seeing at Lenghu site. The DIMM seeing data are collected from October 2018 to December 2020. The histogram is in red and the cumulative probability is in blue.

Line 426, just a comment that the annual temperature variation at Lenghu is remarkably low, particularly the very low range during a day/night cycle.

Line 426, in fig 2, consider separating the data for 2019 and 2020 to make the comparison clearer.

Response:

A good suggestion, thanks. The annual temperature variation at Lenghu is remarkably low, in favor of astronomical observations. We have revised Fig. 2 to display monthly averaged values and standard deviations, and to make better comparison between 2019 and 2020.

Revision in the manuscript:

Fig. 2 | Annual temperature variation pattern in 2019-2020. The upper tip, upper top of the box, mid-bar in the box, bottom of the box and lower tip represent the standard deviation of the maximum temperature, the mean maximum temperature, the mean temperature, the mean minimum temperature, and the standard deviation of the minimum temperature, respectively, at night-time for each month with blue for 2019 and red for 2020. The imbedded panel shows the histogram of the amplitude of the night-time temperature variations.

Line 458, why show lines at 6 and 9 km?

Response:

At an altitude of 6-9 km, the turbulence profile shows a clear difference in August and November, which suggests possible seasonal changes. We added three sentences in the paragraph of ‘Turbulence Profiles’.

Revisions in the manuscript:

(Lines 303-307) Above 11 km, C_N^2 decreases monotonously with no seasonal pattern. C_N^2 is around $10^{-17.5}$ and 10^{-17} between 4 km to 11 km. On November 16, the two turbulence profiles show a similar trend, but the turbulence strength at night (red profile) is lower than that in the morning (grey profile). At an altitude of 6-9 km, the turbulence profile shows a clear difference in August and November, which suggests possible seasonal changes.

D. Appropriate use of statistics and treatment of uncertainties: All error bars should be defined in the corresponding figure legends; please comment if that's not the case. Please include in your report a specific comment on the appropriateness of any statistical tests, and the accuracy of the description of any error bars and probability values.

Yes, all fine, apart from the extraneous sig figs mentioned above.

Response:

Thank you for this comment. We have revised the sig figs in the text. For all figures involve statistics of data, either error bars or quantile or percentile are now given.

E. Conclusions: Do you find that the conclusions and data interpretation are robust, valid and reliable?

Yes.

F. Suggested improvements: Please list additional experiments or data that could help strengthening the work in a revision.

Addressed above.

G. References: Does this manuscript reference previous literature appropriately? If not, what references should be included or excluded?

Yes.

H. Clarity and context: Is the abstract clear, accessible? Are abstract, introduction and conclusions appropriate? Yes to all.

Referee: Michael Ashley, University of New South Wales

Response:

Thank you very much, Michael, for the valuable comments.

-----**Responses to Reviewer #2**-----

Referee #2 (Remarks to the Author):

This paper introduces Lenghu area as a potential reservoir of sites for ground based astronomy in the eastern part of the Tibetan plateau. One of the summit has been thoroughly tested in the past two years, an impressive database of several key parameters has been collected and some results of the analysis are presented. The authors aim at comparing the astroclimate properties of this site to those of a few major reference ground based observatories in operation. They show convincingly that it is in many aspects of comparable quality for future scientific projects. The text is clear and the conclusions are well documented, I consider nevertheless the paper is too modest and should be expanded in some domains to reach the level of expectancy of Nature readers, suggestions are detailed in what follows.

--Introduction

Many readers are not aware of astronomical site selection activities in China, putting the present survey in a global perspective would be helpful: the project should mention in addition to Ali, the other potential sites in the western Tibetan plateau which were part of the site search for the Chinese 12m project. It is obvious, looking at the attached map, that Lenghu had a unique geographical position, filling a gap in the area.

Response:

Thank the reviewer for the comment. Text adjusted. We thank you for providing this map, which is also used to show to the editors and the other referee.

We have added text regarding LOT site surveys and provided quantitative comparison (figure and table) in the Extended Data.

Revisions in the manuscript:

(Lines 151-154) In terms of total seeing, Lenghu is comparable with best established sites (in Chile, Hawaii, and the Canary Islands), and is obviously the best one on the Tibetan Plateau (see Methods, Extended Data Table 2, and Extended Data Figure 4). On the surface of the

earth, the best conditions in both seeing and observing duty cycle for time-domain is clearly on Antarctica^{5,6}.

We added a new section in Methods (Lines 266-278):

Comparison of Tibetan sites: key parameters

In addition to Lenghu, 3 other sites on different locations on the Tibetan Plateau were also tested for different purposes since 2000. An intensive site testing program was carried out for Chinese 12 m telescope recently at these 3 sites, namely Ngari, Muztagh Ata and Daocheng, and are concluded in the overview paper²⁴. It turns out that Lenghu has the best observing condition for optical/infrared. The direct comparisons of the key parameters of AOT and seeing are shown in Extended Data Table 2 and Extended Data Fig. 4, respectively. It is noted that, since the seeing data at Ngari, Muztagh Ata and Daocheng are all truncated at 3.0 arcseconds, the seeing data at Lenghu is also truncated at 3.0 arcseconds in Extended Data Fig. 4.

In Extended Data Table 2, the AOT is calculated using the LOT method based on all sky camera images. They divide the total visible sky by two circles with zenith angle 44.7° and 65°, namely the inner and the outer circle. When there is no cloud in the outer circle, it is defined as ‘clear’ (otherwise photometric in ref. 24); and it is ‘outer’ if only the inner circle is clear (spectroscopic).

Extended Data Table 2. Comparison of key parameters between Tibetan sites.

Site	cAOT*>3 h	AOT*>3 h	25% (arcsec)	50% (arcsec)	75% (arcsec)
Lenghu	73%	84%	0.61	0.75	1.03
Ngari (Ali)	72%	82%	0.88	1.08	1.39
Muztagh Ata	63%	73%	0.64	0.84	1.10
Daocheng	52%	59%	0.82	0.99	1.21

*AOT is the total observing time in a night (photometric and spectroscopic, as in ref. 24), cAOT is contiguous AOT in a night. The rates are the number of nights fulfill the conditions divided by 365.

Extended Data Fig. 4. | Comparison of the cumulative probability of seeing. The red, grey, purple, and blue curves represent the distributions at Lenghu (LH), Muztagh Ata (MA), Daocheng (DC), and Ngari (NG), respectively. Since the seeing data at MA, DC and NG are truncated at 3.0 arcseconds, the seeing data at Lenghu are also truncated at 3.0 arcseconds for uniformity of comparison.

--Water Vapor

Ground based observations in the infrared suffer of the background noise created by the radiation of the telescope structure and components, in particular on "warm" sites like in the coastal Atacama. Hence the very low water vapor period can only be used if the telescope is cooled, either actively which is unrealistic for ELTs, or passively if the site is cold enough. It appears that Lenghu has an enormous advantage in this respect, being the coldest of the Tibetan sites in winter (Ali median is -10C, the 2 others -5C) precisely when the PWV is at the lowest.

Moreover, with global warming predictions of about .5C per decade (attach.2), some of the warmer sites could well reach positive winter temperature within this century.

Response:

The night-time temperature during winter is indeed very low at Lenghu site, compared with other sites. The median night-time temperature during winter (December, January, and February) is -14.5 °C, which is comparable to Ngari and Muztagh Ata, and is obviously

lower than Dao Cheng (Feng et al., 2019). Considering the warming trend of about 0.3 °C per decade (Liu et al., 2017), the night-time temperature at Lenghu site will still be below -10 °C during winter within this century.

Revision in the manuscript:

(Lines 189-193) The median night-time temperature during winter (December, January, and February) is -14.5 °C, which is comparable to Ngari and Muztagh Ata and is significantly lower than Daocheng²⁴. Considering the mean warming trend of about 0.3 °C per decade at Lenghu²⁷, the median night-time temperature in winter will remain below -10 °C toward the end of this century.

References for this response:

24. Feng, L. *et al.* Site testing campaign for the Large Optical/Infrared Telescope of China: Overview. *Res. Astron. Astrophys.*, **20**, 80-94 (2020).
26. Liu, Z., Yang, M., Wan, G., & Wang, X. The spatial and temporal variation of temperature in the Qinghai-Xizang (Tibetan) Plateau during 1971-2015. *Atmos.*, **8**, 214 (2017).

--Seeing

The authors have collected an impressive amount of continuous DIMM (please correct page 3 line 82 DIMM = Differential Image Motion Monitor) data which should allow to extract much more than the median Imn seeing: the median seeing expected to be achieved as a function of the exposure time is also relevant. As it is clear that the Lenghu seeing is variable with a long tail in the distribution, it is important to know if the spikes occur as groups or rather aleatory. An example of such an analysis is given by R. Racine in Mauna Kea in "Temporal fluctuations of atmospheric seeing" 1996, PASP 108, 372-374.

Response:

Very good suggestion, thank you!

We corrected the full name of DIMM. We checked our seeing data according to the method of Racine 1996 and found our seeing data satisfies the log-normal distribution. We also examined the temporal variability of seeing between 1 to 300 minutes. The fraction seeing difference becomes stable at the 250-minute interval with a maximum value of 0.18. The upper panels of Supplementary Figure 3 agree well with the corresponding results from Racine 1996, so the long tail distribution of our seeing is real and is the result of random fluctuations.

Revisions in the manuscript:

(Lines 146-150) We tested temporal variation²⁶ and the wind dependence of seeing and found out that the seeing is stable for most of the observable time. The prevailing wind direction at the site is around 280° throughout the year, which happens to be the wind direction for the best median seeing. The median seeing is below 0.7 arcseconds between wind directions 255° and 324° (see Extended Data Fig. 3).

Extended Data Fig. 3. | Temporal variation and the wind dependence of seeing. **a.** Log-normal distribution of seeing. The maximum number (N) is normalized to 1. **b.** The fraction seeing variation between 1 and 300 minutes according to the method of Racine²⁶. **c.** Seeing versus 2-minute windspeed. **d.** Seeing versus wind direction. In panels **c** and **d**, the solid blue lines are the median while the two blue dotted lines are the 10% and 90% percentiles. The red dashed lines are the number distributions of windspeed and direction when we obtained the seeing measurements. The maximum number is also normalized to 1.

Secondly, it is important to be able to estimate which fraction of the seeing is close to the ground, hence removable on wide field with GLAO technique. The seeing rose, polar plot of the seeing records versus simultaneous incoming wind direction, can help. Should the bad seeing be always linked to the same wind direction, it is clearly caused by local features and thus located close to the surface.

Response:

We plotted seeing versus 2-minute wind speed and wind direction diagrams in the lower panels of Supplementary Figure 1. We found that a smaller wind speed can serve to stabilize the air and make the seeing better. When the wind speed is greater than 5m/s, increased the wind speed starts to induce more turbulence. Seeing is stable at wind speeds between 2 to 8 m/s, which contains 80.5% of our seeing measurements. We did not find a wind direction with bad seeing, but we found that the median seeing was better at wind direction 290°. 34% of our seeing measurements occurred between wind direction 255° and 324°, when the median seeing is smaller than 0.7 arcsec. Both the temporal variation and the wind dependence checks show that Lenghu site seeing is very stable at most observable times.

Revision in the manuscript:

(Lines 140) The seeing measurements obey a log-normal distribution.

(Lines 146-150) We tested temporal variation²⁶ and the wind dependence of seeing and found out that the seeing is stable for most of the observable time. The prevailing wind direction at the site is around 280° throughout the year, which happens to be the wind direction for the best median seeing. The median seeing is below 0.7 arcseconds between wind directions 255° and 324° (see Extended Data Fig. 3).

New reference added:

Racine, R. Temporal Fluctuations of Atmospheric Seeing. *Publ. Astron. Soc. Pac.*, **108**, 372—374 (1996).

New figure added in the Extended Data:

Extended Data Fig. 3. | Temporal variation and the wind dependence of seeing. a. Log-normal distribution of seeing. The maximum number (N) is normalized to 1. **b.** The fraction seeing variation between 1 and 300 minutes according to the method of Racine²⁶. **c.** Seeing versus 2-minute windspeed. **d.** Seeing versus wind direction. In panels **c** and **d**, the solid blue lines are the median while the two blue dotted lines are the 10% and 90% percentiles. The red dashed lines are the number distributions of windspeed and direction when we obtained the seeing measurements. The maximum number is also normalized to 1.

Congratulation to the Lenghu monitoring team for this achievement

Response:

Thank you, again, for the valuable comments.

Reviewer Reports on the First Revision:

Referees' comments:

Referee #1 (Remarks to the Author):

The authors have responded comprehensively to all the comments I made in my earlier review. I particularly appreciated hearing in their rebuttal document about the history of the Tibetan site selection, and the fact that LAMOST is planned to move to Lenghu. Having previously visited Xinglong, and also the Ali site in Tibet, I found this very interesting.

I don't have any additional comments, beyond that I think the article is very worthy of publication in Nature.

Reviewer: Michael Ashley, University of New South Wales

Referee #2 (Remarks to the Author):

I am satisfied with the answers to the questions of the referees.
I agree with the modifications added to the resubmitted paper.

Author Rebuttals to First Revision:

N/A